# Measurement and Estimation of Spectral Sensitivity Functions for Mobile Phone Cameras

**DOI:** 10.3390/s21154985

**Published:** 2021-07-22

**Authors:** Shoji Tominaga, Shogo Nishi, Ryo Ohtera

**Affiliations:** 1Department of Computer Science, Norwegian University of Science and Technology, 2815 Gjøvik, Norway; 2Faculty of Business and Informatics, Nagano University, 658-1, Shimogo, Ueda, Nagano 386-1298, Japan; 3Department of Engineering Informatics, Osaka Electro-Communication University, Neyagawa, Osaka 572-8530, Japan; s-nishi@osakac.ac.jp; 4Kobe Institute of Computing, Graduate School of Information Technology, Chuo-ku, Hyogo, Kobe 650-0001, Japan; ryotera@kic.ac.jp

**Keywords:** mobile phone cameras, spectral sensitivity functions, measurement and estimation, spectral sensitivity database

## Abstract

Mobile phone cameras are often significantly more useful than professional digital single-lens reflex (DSLR) cameras. Knowledge of the camera spectral sensitivity function is important in many fields that make use of images. In this study, methods for measuring and estimating spectral sensitivity functions for mobile phone cameras are developed. In the direct measurement method, the spectral sensitivity at each wavelength is measured using monochromatic light. Although accurate, this method is time-consuming and expensive. The indirect estimation method is based on color samples, in which the spectral sensitivities are estimated from the input data of color samples and the corresponding output RGB values from the camera. We first present an imaging system for direct measurements. A variety of mobile phone cameras are measured using the system to create a database of spectral sensitivity functions. The features of the measured spectral sensitivity functions are then studied using principal component analysis (PCA) and the statistical features of the spectral functions extracted. We next describe a normal method to estimate the spectral sensitivity functions using color samples and point out some drawbacks of the method. A method to solve the estimation problem using the spectral features of the sensitivity functions in addition to the color samples is then proposed. The estimation is stable even when only a small number of spectral features are selected. Finally, the results of the experiments to confirm the feasibility of the proposed method are presented. We establish that our method is excellent in terms of both the data volume of color samples required and the estimation accuracy of the spectral sensitivity functions.

## 1. Introduction

In recent years, mobile phones have become widespread and part of our daily lives (e.g., see [1]). The combination of mobility, telecommunication, and photography enabled by integrating cameras into mobile devices has transformed our lifestyles. In this respect, mobile phone cameras are far more useful than professional digital single-lens reflex (DSLR) cameras. More recently, beyond imaging, mobile sensing technologies using mobile phone cameras have emerged and are rapidly finding applications in many fields, such as smartphone spectroscopy, medical diagnosis, food quality inspection, and environmental monitoring [2,3,4]. The sensing systems often consist of a mobile phone camera and an externally attached device.

Knowledge of the camera spectral sensitivity function is important in many fields that deal with images, such as imaging science and technology, computer vision, medical imaging, and applications involving cultural heritage or artwork. The spectral sensitivity represents the image sensor output per unit incident light energy at each wavelength within the spectral range in which the camera system operates. The function plays the role of mapping the spectral information in a scene to the RGB response values of the camera [5,6]. However, as camera manufacturers typically do not publish this information, users need to either measure or estimate their camera’s sensitivity [7,8]. Thus far, the measurement and estimation of spectral sensitivity functions have been mostly limited to DSLR cameras, for which the digital outputs are available in the form of raw data. For instance, the spectral sensitivity database measured by the Rochester Institute of Technology (RIT) for 28 cameras is given in [9], whereas the Nokia N900 is the only mobile phone (smartphone) for which the sensitivity is given.

Methods for knowing spectral sensitivities can be mainly classified into the direct measurement method and the indirect estimation method. The direct methods are methods in which the spectral sensitivity is measured at each wavelength point in the visible range [7,10]. They require a monochromator that emits stable monochromatic light. Most indirect estimation methods are based on algorithms that use color samples [11,12]. Typical color samples include reflective color targets, such as color checkers, that are photographed by a camera under known illumination. Fluorescence, LED, and LCD display-based color targets can be used as specialized color samples [13,14,15]. The spectral sensitivity functions are estimated from pairs of input and output data comprising the color samples and camera RGB values, respectively. Principal component analysis (PCA) is a combination of direct and indirect methods [16]. This method was described as being more accurate than the conventional indirect method (e.g., see [17]). As estimating the spectral sensitivity of mobile phone camera has not been attempted, we applied the modified method to the spectral sensitivity estimation problem in this study. It should be noted that the spectral sensitivity function creates a linear relationship between the camera inputs and outputs using raw image data.

However, in most cases, the digital output of a mobile phone camera is not raw image data, but rather rendered image data such as JPEG images. This type of data is quite different from raw data as it has undergone many post-processing steps such as white balance (WB), color interpolation, color correction, gamma correction, and compression. Furthermore, the input and output data are not linearly related [17]. Recently, software that can store images captured using a mobile phone camera as raw data has become publicly available. In [18], a compressive sensing approach was proposed for estimating the spectral sensitivity functions of a mobile phone (smartphone) camera. This method is an indirect method based on a limited number of color samples, as the directly measured RGB spectral response functions of the smartphones were not available.

In this study, methods for measuring and estimating the spectral sensitivity functions of mobile phone cameras are developed. Although applying direct measurement methods to mobile cameras gives accurate and reliable results, implementing them over the entire wavelength range of interest in the same manner as professional DSLR cameras is expensive and time-consuming. Indirect estimation methods using color samples face solving high-dimensional matrices of the spectral responses to the samples. The matrices are seriously rank deficient even when the number of color samples is large. Here, we aim to develop an effective estimation method that can achieve sufficient accuracy with a small number of color samples by extracting the features of the spectral sensitivity functions from the dataset of directly measured sensitivities. We use the features of the spectral function shapes, but not the measurements themselves.

In the following, we first describe a direct measurement method for spectral sensitivity functions. An efficient imaging system to generate monochromatic light and measure the camera response is presented. Using this system, we measure a variety of mobile phone cameras available on the market and create a database of spectral sensitivity functions. The directly measured spectral sensitivity functions are used as the reference data to estimate the spectral sensitivity functions.

We then analyze the features of the measured spectral sensitivity functions. The respective spectral sensitivity functions are fitted to color-matching functions for comparison with the spectral sensitivity of the human visual system. PCA is applied to determine the dimensionality of the dataset and to extract the statistical features of the spectral functions.

Subsequently, we present a normal method for estimating spectral sensitivity functions using color samples. We point out the reliability and accuracy drawbacks the normal estimation method suffers from despite being a least squares estimation method. To address this, we propose an effective estimation method using the spectral sensitivity features extracted through PCA in addition to the color samples.

Finally, the results of experiments to confirm the feasibility of the proposed method for estimating the spectral sensitivity functions of a mobile phone camera are presented. We establish that our method has excellent performance in terms of both the data volume of the color samples used and the estimation accuracy of the spectral sensitivities.

## 2. Measurement of Spectral Sensitivity Functions

### 2.1. Measurement Setup

Methodologies for the spectral characterization of RGB cameras have been discussed in some references [19,20]. The standard methods for calibrating consumer cameras in detail, including characteristics such as the linearity, dark current, and spectral response are described in [19]. In this study, we adopted a similar approach to capture images. The camera images were captured in Adobe’s digital negative (DNG) format, which is a lossless raw image format [21]. We manually set the camera International Organization for Standardization (ISO) to 100 and the white balance (WB) mode to incandescent. The exposure time was set to the maximum value at which the dynamic range of the camera outputs did not saturate. The dark response was measured on all the selected cameras by covering the camera with a black sheet in a dark room. The dark response was removed from the camera output. The resulting signal component represents a linear response to the input radiation. The color filter array was a Bayer pattern in which the RGB filters were arranged in a checkerboard pattern with two green pixels (G and G2) for every red or blue one [22]. We averaged the values of the two green pixels. The pixel values for each RGB triplet were averaged over approximately 300 × 300 pixels. The bit depths of the cameras used were 8 to 12 bits.

(a)Linearity

We first evaluated the linearity of the raw camera data. A set of gray chips from the X-rite Color Checker were used as the color target. The surface–spectral reflectance of these samples was measured using a spectral colorimeter (CM-2600d, Konica Minolta, Tokyo, Japan). The mobile phone used was an Apple iPhone 8. Figure 1a shows the relationship between the average reflectance of the gray chips and the camera RGB output. We also calculated the luminance values for the color samples under International Commission on Illumination (CIE) Standard Illuminant D65 using the spectral luminous efficiency curve $\bar{y}(\lambda )$. Figure 1b shows the relationship between the luminance values and the camera RGB outputs.

(b)Spectral response

Figure 2 shows our experimental setup for measuring the spectral responses of mobile phone cameras using monochromatic light and a spectrometer [23]. Figure 2a shows the conversion of the continuous spectrum from a xenon lamp into monochromatic light using the grating in a monochromator (SPG-100, Shimazu Kyoto, Japan). The monochromatic light was projected onto a diffuser, and the transmitted light image was observed using both a mobile phone camera and a spectroradiometer (CS-2000, Konica Minolta, Tokyo, Japan). The effective spectral resolution (full width at half maximum (FWHM)) of the monochromator was approximately 4 nm.

Figure 2b shows the production of monochromatic light by a programmable light source (OL490, Optronic Laboratories, Orlando, FL, USA) and a liquid light guide. The lighting system was a spectrally controllable light source using a digital micromirror device (DMD) [24]. This lighting system was used in this study to generate emissions with a narrow width at a single wavelength in the visible range (400–700 nm). The emitted light was projected onto a white reference standard (Spectralon), and the reflected light image was observed by the camera and spectroradiometer. The FWHM was approximately 5–10 nm. We used mainly the system shown in Figure 2b for the measurement. When the monochromator system was used, the average values of the measured spectral sensitivity functions for both systems in Figure 2a,b were taken.

Assuming linear camera response, the three-channel output of a mobile phone camera can be described as
(1)$$\left[\begin{array}{c} R\\ G\\ B \end{array}\right]=\int_{380}^{720}e\left(\lambda \right)\left[\begin{array}{c} r(\lambda )\\ g(\lambda )\\ b(\lambda ) \end{array}\right]d\lambda ,$$
where *R*, *G*, and *B* represent the camera responses after reducing the dark responses. e(λ) represents the illuminant spectrum and r(λ), g(λ), and b(λ) the spectral sensitivity functions. We denote the spectral powers of the *n* illuminants used in the measurement as
(2)$$\int_{380}^{720}e_{i}\left(\lambda \right)d\lambda \;=\;\;E_{i},\;\;\;\;\;\;\;(i=1,\;2,\;\ldots ,\;n).$$

When each illuminant spectrum is unimodal, such as monochromatic light, and the FWHM is sufficiently narrow compared to the sensitivity functions, the spectral sensitivity at each wavelength
$\lambda_{i}$ can be calculated from the illuminant power and the camera outputs as
(3)$$\left[\begin{array}{c} r(\lambda_{i})\\ g(\lambda_{i})\\ b(\lambda_{i}) \end{array}\right]=\left[\begin{array}{c} R_{i}/E_{i}\\ G_{i}/E_{i}\\ B_{i}/E_{i} \end{array}\right],\;\;\;\;\;\;\;(i=1,\;2,\;\ldots ,\;n).$$
where $r(\lambda_{i})$, $g(\lambda_{i})$, and $b(\lambda_{i})$ (*i* = 1, 2, ..., *n*) are the discrete representations of the spectral sensitivity functions. The visible wavelength range (400–700 nm) was scanned at equal wavelength intervals of 10 nm and each spectral sensitivity function was expressed as a 31-dimensional column vector with *n* = 31.

### 2.2. Spectral Sensitivity Database

We measured the spectral sensitivities of 20 mobile phone cameras and constructed a database of the spectral sensitivity functions. The mobile phones used in this study are listed in Table 1. Typically, the mobile phone incorporates an image sensor produced by a different manufacturer. The third column in Table 1 lists the names of the image sensors. Figure 3 shows the relative spectral sensitivity functions of the 20 mobile phone cameras in our database. Most mobile phones are based on either iOS or Android. Six of the mobile phones in Table 1 are iOS phones and the remaining 14 are Android phones. Figure 4 shows the spectral sensitivity functions grouped into the two categories of (a) iOS phone cameras and (b) Android phone cameras. There is no significant difference in the spectral curves of the sensitivity functions between the two categories.

It may be interesting to compare this dataset with a dataset of the DSLR cameras. Compared to the spectral sensitivity database in [9], we see that the shapes of the spectral distributions in our mobile phone cameras do not have large variations, differently from those of the DSLRs.

The numerical data of the spectral sensitivity functions in Table 1 have been published as Excel data and txt data on http://ohlab.kic.ac.jp/ (accessed on 22 July 2021).

## 3. Feature Analysis of Spectral Sensitivity Functions

### 3.1. Fitting to Color-Matching Functions

The Luther condition states that the camera spectral sensitivity functions are linear transformations of the CIE-1931 2-degree color-matching functions [25]. As the color-matching functions are a numerical representation of the color vision response of the standard observers, the degree of fit with the Luther condition determines the colorimetric measurement accuracy of the mobile phone cameras.

Let $[\bar{x},\;\bar{y},\;\bar{z}]\;$ be the 31 × 3 discrete matrix representation of the CIE-1931 2-degree color-matching functions $\bar{x}(\lambda )$, $\bar{y}(\lambda )$, $\bar{z}(\lambda )$ and [**r**, **g**, **b**] be the 31 × 3 matrix representing the RGB spectral sensitivity functions of a mobile phone camera. We express the linear relationship between these matrices as
(4)$$[\bar{x},\;\bar{y},\;\bar{z}]\;\;=\;[r,\;g,\;b]T,$$
where **T** is a 3
× 3 transformation matrix. To validate whether the measured sensitivity functions satisfy the Luther condition, we estimate the matrix **T** using the least squares solution for Equation (4). The estimate is given as
(5)$$\hat{T}={\left[CC^{t}\right]}^{-1}C^{t}A,$$
where $A=[\bar{x},\;\bar{y},\;\bar{z}]\;$, C= [r, g, b], and the symbol *t* denotes matrix transposition.

The root-mean-square errors (RMSEs) $\sqrt{trace{\left[A-C\hat{T}\right]}^{t}\left[A-C\hat{T}\right]/(31\times 3)}$ of the estimates were calculated, where the symbol trace[X] denotes the trace of a matrix X. Figure 5 shows the fitting results to the color-matching functions based on the spectral sensitivity functions of the iPhone 8. The RMSE was 0.254. The average RMSE over all the measured sensitivity functions in our database was 0.226. We also calculated the CIE-LAB color difference (Delta-E 1976) under Standard Illuminant D65 for many Munsell color chips. The average color difference was 5.61. Thus, the Luther condition was not completely satisfied.

### 3.2. PCA Analysis

PCA was applied to the measured spectral function database to determine the dimensionality for approximating the spectral data and statistically extracting the spectral shape features. All the relative spectral curves of the measured spectral sensitivity functions from the *N* (=20) mobile phone cameras are represented for each channel in a 31 × *N* matrix as
(6)$$Y_{R}\;=\;[r_{1},\;r_{2},\;\ldots ,\;r_{N}],\;\;\;\;\;\;\;\;Y_{G}\;=\;[g_{1},\;g_{2},\;\ldots ,\;g_{N}],\;\;\;\;\;\;\;Y_{B}\;=\;[b_{1},\;b_{2},\;\ldots ,\;b_{N}].$$

We summarize these matrices as
(7)$$Y_{k}\;=\;[y_{1},\;y_{2},\;\ldots ,\;y_{N}]\;\;\;\;\;\;\;\;\;\;\;\;\;\;\;\;\;\;\;\;\;\;(k=R,\;G,\;B).$$

Singular-value decomposition (SVD) provides an orthogonal decomposition of rectangular matrices [26]. We used SVD for the PCA. The SVD of each matrix $Y_{k}$ is written as
(8)$$Y_{k}\;=\;U\Sigma V^{t}$$
or equivalently,
(9)$$Y_{k}\;=\;\sigma_{1}u_{1}v_{1}{}^{t}+\sigma_{2}u_{2}v_{2}{}^{t}+\;\ldots +\sigma_{N}u_{N}v_{N}{}^{t},$$
where $U\;(=[u_{1},\;u_{2},\;\ldots \;,u_{N}])$ and $V\;(=[v_{1},\;v_{2},\;\ldots \;,v_{N}])$ are the 31 × *N* left singular matrix and the *N*
× *N* right singular matrix, respectively, and Σ is the *N*
× *N* diagonal matrix in which the elements are the singular values $\sigma_{1},\;\sigma_{2},\;\ldots ,\;\sigma_{N}\;\;(\sigma_{i}\geq \sigma_{i+1}>0)$. The 31-dimensional singular vectors $u_{1},\;u_{2},\;\ldots \;,u_{N}$ are orthogonal to each other. Therefore, each measured spectral sensitivity function can be uniquely expressed as a linear combination of *N* orthogonal vectors
(10)$$y_{j}\;=\;c_{1j}u_{1}+c_{2j}u_{2}\;\ldots +c_{Nj}u_{N},\;\;\;\;\;\;\;\;\;\;\;(j=1,\;2,\;\ldots ,\;N)$$
where $c_{ij}=\sigma_{i}v_{ij}$. Consider an approximate representation of the sensitivity functions in terms of component vectors chosen from $u_{1},\;u_{2},\;\ldots \;,u_{N}$. When the first *K* principal components are chosen, the performance index representing the approximation accuracy is given by the percent variance:(11)$$P(K)=\sum_{i=1}^{K}\sigma_{i}^{2}/\sum_{i=1}^{N}\sigma_{i}^{2}.$$

Figure 6 shows the first three principal components $u_{1}$, $u_{2}$, and $u_{3}$ for the (a) red, (b) green, and (c) blue channels of the spectral sensitivity functions where the bold, broken, and dotted curves represent the first, second, and third principal components, respectively. The percent variances are *P*(1) = 0.9940, *P*(2) = 0.9974, *P*(3) = 0.9990 for red, *P*(1) = 0.9954, *P*(2) = 0.9984, *P*(3) = 0.9995 for green, and *P*(1) = 0.9926, *P*(2) = 0.9962, *P*(3) = 0.9999 for blue. The first principal component $u_{1}$ is the average spectral curve of the spectral sensitivity function dataset, which plays the most important role in the spectral representation. The results of the PCA suggest that the measured spectral sensitivity functions can be approximated using the first three principal components with sufficient accuracy. The approximation is obtained using the principal component expansion in Equation (10). Figure 7 shows the approximated spectral curves for the (a) red, (b) green, and (c) blue channels of the spectral sensitivity functions of the iPhone 8, where the colored bold curves, the black bold curves, and the broken curves represent the measured spectral sensitivities, the approximation using the first component only, and the approximation using the first two components, respectively.

## 4. Estimation of Spectral Sensitivity Functions

### 4.1. Normal Method Using Color Samples

The direct measurement method for the spectral sensitivity functions of mobile phone cameras is time-consuming and expensive, and requires a laboratory setting. Therefore, despite their inferior accuracy, indirect estimation methods are often used. The indirect methods are normally based on color samples. The spectral sensitivity functions are estimated from a pair comprising the input data of the color samples and the corresponding output data of the captured RGB values.

Suppose that *M* different color samples with surface spectral reflectance $S_{i}\left(\lambda \right)$ (*i* = 1, 2, …, *M*) are observed under illumination e(λ) by a mobile phone camera. The camera inputs are the spectral data $S_{i}\left(\lambda \right)e(\lambda )$ reflected from the color samples, which are summarized as an *n*
× *M* matrix
(12)$$C=[s_{1}.*e,\;\;s_{2}.*e,\;\;\ldots ,\;\;s_{M}.*e],$$
where $s_{i}$ (*i* = 1, 2, ..., *M*) and e are *n*-dimensional column vectors representing the spectral reflectances and the illuminant, respectively, and the symbol. * represents elementwise multiplication. The camera outputs for the *M* color samples are represented as the 1 × *M* matrices
(13)$$z_{R}=[R_{1},\;R_{2},\;\ldots ,\;R_{M}],\;\;\;\;z_{G}=[G_{1},\;G_{2},\;\ldots ,\;G_{M}],\;\;\;\;\;z_{B}=[B_{1},\;B_{2},\;\ldots ,\;B_{M}].$$

The observed outputs can then be written in the discrete form
(14)$$z_{R}=r^{t}C,\;\;\;\;z_{G}=g^{t}C,\;\;\;\;\;z_{B}=b^{t}C,$$

The least squares estimates for Equation (14) are given by
(15)$$\hat{r}={\left[CC^{t}\right]}^{-1}Cz_{R}{}^{t},\;\;\;\;\hat{g}={\left[CC^{t}\right]}^{-1}Cz_{G}{}^{t},\;\;\;\;\;\hat{b}={\left[CC^{t}\right]}^{-1}Cz_{B}{}^{t}.$$

However, this normal method is often ineffective for accurately estimating the spectral sensitivity functions owing to the dimensionality of the surface spectral reflectance. As the surface–spectral reflectance of natural or human-made objects are described using six to eight basis functions [27,28], the dimension of the reflectance is much lower than the dimension *n* (=31) of the spectral sensitivity functions. Therefore, even when many samples *M* > *n* are used, the matrix **C** may be rank deficient and the matrix inversion is often unreliable.

### 4.2. Proposed Method Based on Color Samples and Spectral Features

We consider solving the estimation problem by using the spectral features of the sensitivity functions obtained through PCA in addition to the color samples. We focus on a linear model representation of the spectral sensitivity functions. The PCA results in the previous section suggest that the spectral sensitivity functions of mobile phone cameras can be approximated by linear combinations of a small number of principal components. Selecting the first *L* components for the estimates, the linear model is represented as
(16)$$r=\sum_{i=1}^{L}x_{Ri}u_{Ri},\;\;\;\;\;\;g=\sum_{i=1}^{L}x_{Gi}u_{Gi},\;\;\;\;\;\;b=\sum_{i=1}^{L}x_{Bi}u_{Bi},$$
where $u_{Ri}$, $u_{Gi}$, and $u_{Bi}$ are the principal component vectors for each of the red, green, and blue channels, respectively, as shown in Figure 6, and the coefficients $x_{Ri}$, $x_{Gi}$, and $x_{Bi}$ are unknown scalar weights to be estimated. Equation (16) provides a strong constraint in estimating the spectral shapes of the sensitivity functions.

Writing the *L* principal components of the spectral sensitivity features and the corresponding weights as the *n*
× *L* matrices
(17)$$U_{R}=\left[u_{R1},\;\ldots ,\;u_{RL}\right],\;U_{G}=\left[u_{G1},\;\ldots ,\;u_{GL}\right],\;U_{B}=\left[u_{B1},\;\ldots ,\;u_{BL}\right],$$
and *L*-dimensional column vectors
(18)$$x_{R}={\left[x_{R1},\;\ldots ,\;x_{RL}\right]}^{t},\;x_{G}={\left[x_{G1},\;\ldots ,\;x_{GL}\right]}^{t},\;x_{B}={\left[x_{B1},\;\ldots ,\;x_{BL}\right]}^{t},$$

We have a compact form for Equation (16):(19)$$r=U_{R}x_{R},\;\;\;\;\;g=U_{G}x_{G},\;\;\;\;\;b=U_{B}x_{B}.$$

The spectral features of r, g, and b are included in the arrays of $U_{R}$, $U_{G}$, and $U_{B}$.

The next step is to determine the weighting coefficients. Substituting Equation (19) into Equation (14), we obtain the following observation equations for the weights
(20)$$z_{R}=x_{R}{}^{t}U_{R}{}^{t}C,\;\;\;\;\;z_{G}=x_{G}{}^{t}U_{G}{}^{t}C,\;\;\;\;\;\;\;z_{B}=x_{B}{}^{t}U_{B}{}^{t}C.$$

Therefore, the weights $x_{R}$, $x_{G}$, and $x_{B}$ can be estimated using the observed dataset of *M* color chips$z_{R}$, $z_{G}$, and $z_{B}$. The least-squares estimates are given as
(21)$${\hat{x}}_{R}={\left[C_{R}C_{R}{}^{t}\right]}^{-1}C_{R}z_{R}{}^{t},\;\;\;\;{\hat{x}}_{G}={\left[C_{G}C_{G}{}^{t}\right]}^{-1}C_{G}z_{G}{}^{t},\;\;\;\;\;{\hat{x}}_{B}={\left[C_{B}C_{B}{}^{t}\right]}^{-1}C_{B}z_{B}{}^{t}$$
where $C_{R}=U_{R}{}^{t}C$, $C_{G}=U_{G}{}^{t}C$, and $C_{B}=U_{B}{}^{t}C$. We note that the matrix sizes of $C_{R}$, $C_{G}$, and $C_{B}$ are *L*
× *M*, and $C_{R}C_{R}{}^{t}$, $C_{G}C_{G}{}^{t}$, and $C_{B}C_{B}{}^{t}$ are full-rank *L*
× *L* matrices. Therefore, the matrix inversion is stable. The estimates of the spectral sensitivity functions $\hat{r}$, $\hat{g}$, and $\hat{b}$ are finally obtained by substituting ${\hat{x}}_{R}$, ${\hat{x}}_{G}$, and ${\hat{x}}_{B}$ into Equation (19).
(22)$$\hat{r}=U_{R}{\left[C_{R}C_{R}{}^{t}\right]}^{-1}C_{R}z_{R}{}^{t},\;\;\;\;\hat{g}=U_{G}{\left[C_{G}C_{G}{}^{t}\right]}^{-1}C_{G}z_{G}{}^{t},\;\;\;\;\;\hat{b}=U_{B}{\left[C_{B}C_{B}{}^{t}\right]}^{-1}C_{B}z_{B}{}^{t}.$$

It should be noted that these estimates are the least squares estimates to minimize the residual error in the observations. The original residual error is described by a cost function $L_{F}$ as
(23)$$L_{F}={\left\|z_{R}-r^{t}C\right\|}^{2}+{\left\|z_{G}-g^{t}C\right\|}^{2}+{\left\|z_{B}-b^{t}C\right\|}^{2}$$
where the symbol ${\left\|x\right\|}^{2}$ denotes the L2 norm of a vector x. As the spectral sensitivity functions are represented using the linear model in Equation (16), the cost function can be rewritten as follows:(24)$$L_{F}={\left\|z_{R}-x_{R}^{}{}^{t}U_{R}{}^{t}C\right\|}^{2}+{\left\|z_{G}-x_{G}^{}{}^{t}U_{G}{}^{t}C\right\|}^{2}+{\left\|z_{B}-x_{B}^{}{}^{t}U_{B}{}^{t}C\right\|}^{2}$$

When *L* is directly minimized with respect to each of $x_{R}$, $x_{G}$, and $x_{B}$, we can obtain the same solution as Equation (22). Therefore, the proposed method solves the optimization problem to minimize the residual error.

The parameter *L* denotes the dimensionality of the linear model representation of the spectral sensitivity functions. The selection of *L* affects the estimation accuracy. The most appropriate value of *L* is determined experimentally based on the estimation accuracy of the spectral sensitivity functions.

## 5. Experimental Results

### 5.1. Experimental Setup

We used 1523 Munsell color chips from the Munsell Book of Color [29]. The chips are reflective color targets arranged in the Munsell color system. Figure 8 shows the imaging setup for the color samples using a mobile phone camera. We measured the surface–spectral reflectance of each color sample using the spectral colorimeter. Figure 9 shows the complete set of all the spectral reflectance measurements. We also used 24 color samples from the X-Rite ColorChecker Passport Photo [30]. Figure 10 shows the color checkers used for the validation of sensitivity measurement and reflectance estimation, where panel (a) shows the imaging targets consisting of 24 color checkers and the white reference standard (Spectralon), and panel (b) the spectral reflectance values measured by the spectral colorimeter. The illumination light source was an incandescent lamp with the spectral power distribution shown in Figure 11. A Munsell white paper (N9.5) was used to correct the illumination unevenness on the color samples and limb darkening.

### 5.2. Validation of the Measured Spectral Sensitivities

To evaluate the accuracy of the measured spectral sensitivities, the color differences were computed between the colors imaged by the mobile phone cameras and the simulated colors based on the measured spectral sensitivities. We randomly selected two iOS and Android phone cameras, making a total of four phone cameras, which we studied using the X-Rite color checkers in Figure 10.

We simulated the camera response for each color checker using the measured spectral sensitivities $\left(r_{\mathrm{mea}},\;g_{\mathrm{mea}},b_{\mathrm{mea}}\right)$ of each mobile phone camera. The camera output RGB values $\left(z_{\mathrm{mea},\;R},\;z_{\mathrm{mea},\;G},\;z_{\mathrm{mea},\;B}\right)$ can be predicted in a discrete form as
(25)$$z_{\mathrm{mea},\;R}=r_{\mathrm{mea}}^{t}s.*e,\;\;\;\;\;\;z_{\mathrm{mea},\;G}=g_{\mathrm{mea}}^{t}s.*e,\;\;\;\;\;\;\;z_{\mathrm{mea},\;B}=b_{\mathrm{mea}}^{t}s.*e$$
where s and e represent the spectral reflectance of the color checker in Figure 10b and the illuminant spectral power distribution in Figure 11, respectively.

On the other hand, the RGB values $\left(z_{\mathrm{img},\;R},\;z_{\mathrm{img},\;G},\;z_{\mathrm{img},\;B}\right)$ of the color checkers were captured directly by each camera. The RGB color difference was then averaged over all color checkers as
(26)$$\Delta E_{\mathrm{RGB}}\;=E\left[\sqrt{{\left(z_{\mathrm{img},\;R}-z_{\mathrm{mea},\;R}\right)}^{2}+{\left(z_{\mathrm{img},\;G}-z_{\mathrm{mea},\;G}\right)}^{2}+{\left(z_{\mathrm{img},\;B}-z_{\mathrm{mea},\;B}\right)}^{2}}\right]$$

Table 2 shows the average color differences between the imaged colors of the 24 color checkers captured by each camera and the simulated colors based on the measured spectral sensitivities, where the RGB values are normalized to be $\sqrt{R^{2}+G^{2}+B^{2}}=1$ for the white reference standard under the incandescent lamp used.

### 5.3. Estimation Results by the Normal Method

The estimation results obtained using the normal method for the iPhone 8 are shown in Figure 12, where the bold curves in red, green, and blue represent the estimated spectral sensitivities of the red, green, and blue channels, respectively, and the broken curves represent the measured spectral sensitivities. A non-negative least squares estimation was applied to solve Equation (14). Despite the use of many color samples, the estimation results based on the direct use of color samples were quite unstable and inaccurate.

To investigate the dimensionality of the spectral reflectance dataset, SVD was applied to the 31 × 1523 matrix $S=[s_{1},\;s_{2},\;\ldots ,\;s_{1523}]$ consisting of the measured spectral reflectance values from the Munsell color chips. Figure 13 depicts the percent variance P(K) defined in Equation (11) as a function of the number of principal components *K*. It can be seen that the percent variance reached 1.0 around *K* = 6. Therefore, the dimension of the reflectance dataset was approximately 6 and it was not possible to recover the spectral functions with 31 dimensions.

### 5.4. Estimation Results by the Proposed Method

The feasibility of the proposed method was evaluated in two steps.

(1) In the first experiment, we focused solely on the iPhone 8 and investigated the performance in detail. The spectral sensitivity functions of the iPhone 8 were removed from the original database, and SVD was applied to the remaining dataset to obtain the principal components $u_{Ri}$, $u_{Gi}$, and $u_{Bi}$. Next, the spectral sensitivities of the iPhone 8 were estimated via the algorithm in Section 4.2 using the camera RGB values for the color samples and the principal components. Finally, to determine the most appropriate *L*, the root mean squared errors of the estimated spectral sensitivities were calculated as
(27)$$rmse=\;\sqrt{\left({\left\|\hat{r}-r\right\|}^{2}+{\left\|\hat{g}\;-g\right\|}^{2}+{\left\|\hat{b}-b\right\|}^{2}\right)/\left(31\times 3\right)}.$$

The estimation errors at different values of *L* were *rmse* = 0.04929, 0.07239, and 0.08963 for *L* = 1, 2, and 3, respectively. The minimum error was recorded when the estimation was performed using only one component. Figure 14 shows the estimated spectral sensitivity functions of the iPhone 8 at *L* = 1.

Furthermore, we examined the possibility of reducing the number of color samples. Suppose that *m* color samples are used for sensitivity estimation. We randomly selected *m* samples from all the 1523 samples of the Munsell color chips and then computed the spectral sensitivity estimate and accuracy based on the selected sample data. This trial was repeated 1000 times to achieve significant statistical reliability. The average errors were *rmse* = 0.04958, 0.04933, 0.04931, 0.04929, and 0.04929 for *m* = 10, 50, 100, 500, and 1000. These results show that a sufficient estimation accuracy can be obtained even when only ten color samples are used. Figure 15 shows the estimated sensitivities of the iPhone 8 at *L* = 1 and *m* = 10. Thus, the proposed method has the advantage that the spectral sensitivity functions can be estimated using the average spectral curves of the database and a small number of color samples.

In the second experiment, we investigated whether the performance of the proposed method depends on the mobile phone cameras used and the color samples. As the measured spectral sensitivity curves were similar, we selected two iOS phone cameras and two Android phone cameras, and evaluated their spectral sensitivities using the 24 color sample from the X-Rite color checkers. For fair validation, the principal components were computed using the dataset of the spectral sensitivity measurement for the remaining 19 mobile phone cameras except for the target phone camera. Table 3 lists the RMSEs of the estimated spectral sensitivities for the four mobile phone cameras for the principal components with different *L* values. As can be seen from the table, the minimum errors were found when the first component in all cases of the four mobile phone cameras was set at *L* = 1. Figure 16 depicts the estimated sensitivity functions of the four cameras at *L* = 1. Thus, it can be confirmed that the spectral sensitivity of the mobile phone camera can be estimated based solely on the first principal component of the dataset and a small number of color samples.

### 5.5. Reflectance Estimation Validation

The surface–spectral reflectance provides is a physical feature inherent to the surface of a target object. Therefore, estimating the spectral reflectance is crucial for object recognition and identification in a variety of fields, including image science and technology, and computer vision. To evaluate the feasibility of mobile phones for spectral reflectance estimation, we validated the measured and estimated spectral sensitivity functions of the iPhone 8.

The Wiener filter is a linear estimation technique widely used for spectral reflectance estimation [31,32,33,34,35,36,37]. The relationship between the spectral reflectance of an object surface and the camera outputs is modeled with the camera output RGB vector $z={\left[z_{R},\;z_{G},\;z_{B}\right]}^{t}$, the spectral sensitivity vectors **r**, **g**, and **b**, the illuminant vector **e**, the reflectance vector **s** to be estimated, and the noise vector **n** in observation as
(28)z = As + n.
where $A\;={\left[r.*e,\;g.*e,\;b.*e\right]}^{t}$. When the reflectance **s** and noise **n** are uncorrelated, the Wiener estimate with the minimal mean square error is expressed as follows:(29)$$\hat{s}\;=\;PA^{t}{\left(APA^{t}\;+\;\Sigma \right)}^{-1}z.$$
where **P** is the covariance matrix of the reflectance data and Σ the covariance matrix of the noise, which can usually be assumed to be a diagonal matrix $\Sigma =\mathrm{diag}(\sigma_{R}^{2},\;\sigma_{G}^{2},\;\sigma_{B}^{2})$. We determined **P** using the database of surface spectral reflectance values in [38] and determined Σ empirically (see [38]).

The accuracy of the estimated reflectance values was evaluated using the average RMSE and the average CIE-LAB color difference obtained over the 24 color checkers. Table 4 lists the performance values of the iPhone 8 in four cases where measurements 1 and 2, respectively, are based on the directly measured spectral sensitivities by the monochromator in Figure 2a and the measured spectral sensitivities by the programmable light source in Figure 2b, and Estimations 1 and 2, respectively, are based on the estimated spectral sensitivities from all the color samples in Figure 14 and the estimated spectral sensitivities using only 10 color samples in Figure 15. The LAB color difference was calculated using the illuminant shown in Figure 11. There is no noticeable difference between the performances of the four spectral sensitivity functions used in the validation. The validation results suggest that both the measured and the estimated spectral sensitivities in this study are useful for spectral reflectance estimation.

## 6. Discussion

It is important to clarify the difference between the estimation accuracy and the approximation accuracy. This distinction can be drawn when the spectral sensitivities are known. In the proposed method, the spectral sensitivities of the mobile phone cameras are estimated in a linear combination of the principal components of the dataset, and approximated in a linear combination of the same principal components. When using the first *K* components, the error between the original spectral sensitivity matrix Y and the approximated matrix ${\hat{Y}}_{K}$ can be expressed as follows:(30)$${\left\|Y-{\hat{Y}}_{K}\right\|}^{2}=\;\sigma_{K+1}^{2}+\sigma_{K+2}^{2}+\cdots +\sigma_{N}^{2}$$
where $\sigma_{1},\;\sigma_{2},\;\ldots ,\;\sigma_{N}\;$are singular values (see Equation (9)). Therefore, the approximation error monotonically decreases as the component number *K* increases. However, the behavior of the estimation accuracy is not necessarily the same. The larger the component number, the smaller the singular values and the lower the contribution rate. In such a situation, when a component with a small contribution that is regarded as noise is added to the estimation process, the estimation accuracy deteriorates extremely. Figure 17 shows the variations in the estimation and approximation errors as a function of the principal components. The error values are averaged over the four mobile phone cameras in Table 3. The minimum average estimation error is recorded when only one principal component is used, and the second smallest error is realized when three principal components are used. As the contribution rate of the first principal component is very large as *P* (1) > 0.99, and in contrast the contribution of the third principal component is significantly smaller, the estimation using only the first principal component can be considered stable and reliable.

Based on the experimental results, it was found that the spectral sensitivity functions of the mobile phone cameras can be estimated from only the first principal component of the dataset of the measured spectral sensitivities and a small number of color samples. In the case of DSLR cameras, two principal components were required and the extent to which the color samples could be simplified was not clear [16]. The present results suggest that the spectral sensitivity functions of the mobile phone cameras can be more easily estimated using from 10 to 24 color samples, compared with DSLR cameras. From a computational simplicity point of view, we note that the inverse matrix operation is not required because the matrices $C_{i}C_{i}{}^{t}$ (*i* = R, G, B) become scalar values. As a result, the spectral sensitivity functions can be modeled using the first principal components as follows:(31)$$\hat{r}=c_{R}u_{R},\;\;\;\;\hat{g}=c_{G}u_{G},\;\;\;\;\;\hat{b}=c_{B}u_{B}.$$

As the first principal component is the same as the average spectrum of the spectral dataset, the PCA of the dataset of the measured spectral sensitivities is not required. The spectral sensitivity functions are estimated by simply weighting the average spectral sensitivity functions. The numerical data of the average spectral sensitivities $u_{R}$, $u_{G}$, and $u_{B}$ are available at the same site (http://ohlab.kic.ac.jp/) (accessed on 22 July 2021) as our database. The three weighting coefficients $c_{R}$, $c_{G}$, and $c_{B}$ are calculated from the camera RGB data for color samples and the estimates of the spectral sensitivity functions are obtained using Equation (31).

## 7. Conclusions

We have developed methods for measuring and estimating the spectral sensitivity functions of mobile phone cameras. In the direct measurement method, the spectral sensitivity at each wavelength was measured using monochromatic light. Although this method was accurate, it is time-consuming and expensive. The indirect estimation method was based on color samples from which the spectral sensitivities were estimated using pairs of input and output data comprising the color samples and the corresponding camera output RGB values, respectively.

We first presented an imaging system for direct measurements and performed measurements on a variety of mobile phone cameras to create a database of spectral sensitivity functions. Subsequently, the features of the measured spectral sensitivity functions in our database were analyzed using PCA. We determined the dimensionality of the spectral sensitivity dataset and extracted the statistical features of the spectral functions. We then described a normal method for estimating the spectral sensitivity functions using color samples. However, this method was not effective at estimating the spectral sensitivity functions. Therefore, we proposed an effective method to solve the estimation problem using the spectral features of the sensitivity functions in addition to the color samples. The estimation was accurate even when only a small number of spectral features were selected.

The feasibility of the proposed method was confirmed through experiments. The characteristics and advantages of the proposed method over the previously published methods for mobile phone cameras are summarized as follows:

1. We measured the spectral sensitivities of mobile phone cameras, created a database, and clarified its characteristics for the first time.

2. The spectral sensitivities can be estimated using the average spectral sensitivity and a small number of color samples.

3. The computation is stable and straightforward because of determining only three unknown parameters, and the estimation accuracy is very close to the direct measurement results.

The limitation of our approach is that it does not have the manufacturers’ data to validate the proposed method properly. However, with direct measurements, using PCA provides a means to optimally approximate the sensitivities’ spectral power distribution, which helps us overcome the limitations of our approach, even in short of the manufacturer‘s data.

The number of people using DSLR cameras, other than professional photographers, is decreasing worldwide, and mobile phones are becoming the mainstream. The advantages of mobile phone cameras include portability, low cost, convenience, and a wide range of applications. Unlike DSLR cameras, the camera and computer come together as a set in a mobile phone. This is advantageous for implementing applications that require the use of spectral sensitivity functions. We believe that this paper adds value to many fields including imaging science and technology as the techniques to estimate the spectral sensitivities of mobile phone cameras are still developing. Furthermore, our study makes an effective contribution to show journal readers different approaches, as there is intense market growth in mobile phones.

## Figures and Tables

**Figure 1 sensors-21-04985-f001:**
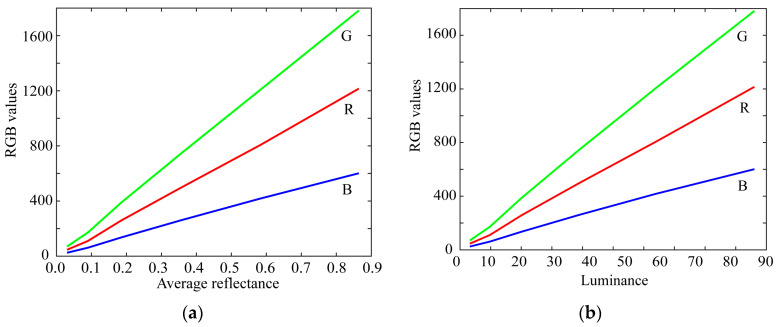
Linearity test of the raw camera data using color samples. (**a**) Relationship between the average reflectance of gray chips and the camera RGB outputs. (**b**) Relationship between the luminance values and the camera RGB outputs.

**Figure 2 sensors-21-04985-f002:**
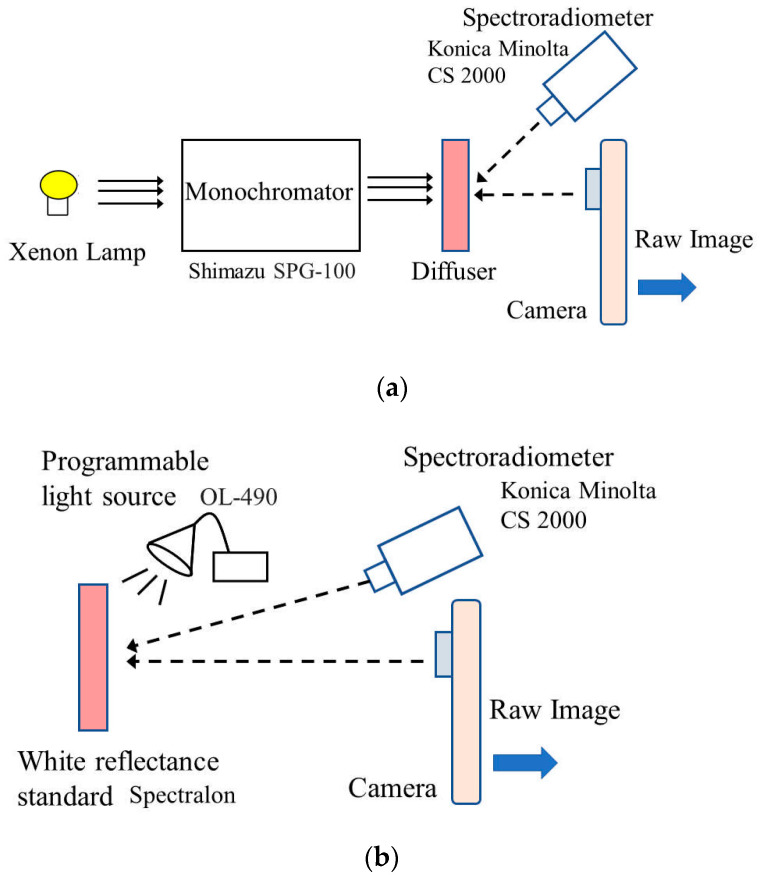
Experimental setups for measuring the spectral responses of mobile phone cameras using monochromatic light and a spectrometer. (**a**) Monochromatic light from monochromator grating. (**b**) Monochromatic light from a programmable light source.

**Figure 3 sensors-21-04985-f003:**
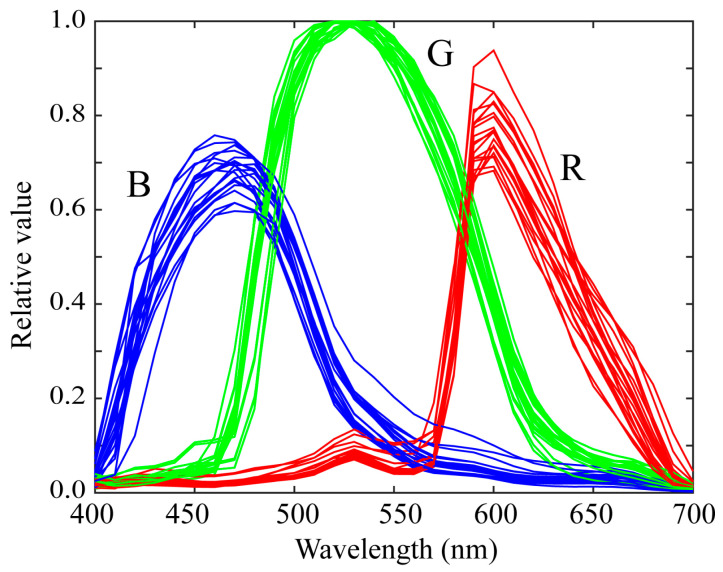
Relative spectral sensitivity functions of the 20 mobile phone cameras in our database.

**Figure 4 sensors-21-04985-f004:**
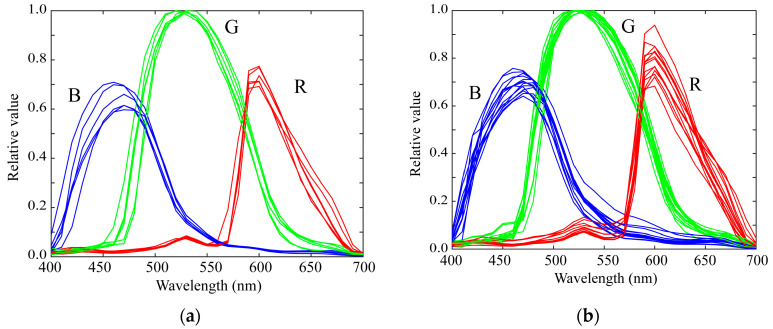
Spectral sensitivity functions classified into the two categories of (**a**) iOS phone cameras, and (**b**) Android phone cameras.

**Figure 5 sensors-21-04985-f005:**
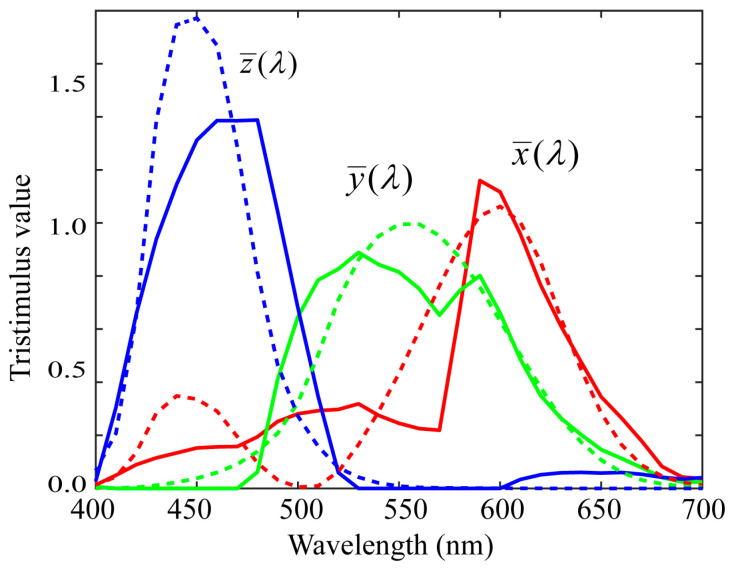
Fitting results to the color-matching functions based on the spectral sensitivity functions of iPhone 8.

**Figure 6 sensors-21-04985-f006:**
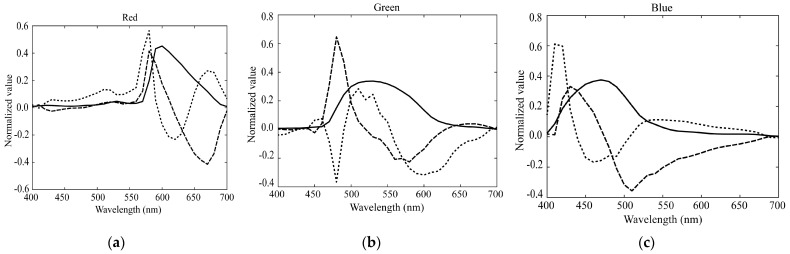
First three principal components $u_{1}$, $u_{2}$, and $u_{3}$ for the (**a**) red, (**b**) green, and (**c**) blue channels of the spectral sensitivity functions. The bold, broken, and dotted curves represent the first, second, and third principal components, respectively.

**Figure 7 sensors-21-04985-f007:**
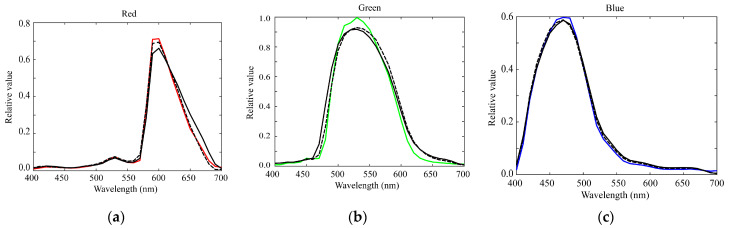
Approximated spectral curves for the (**a**) red, (**b**) green, and (**c**) blue channels of the iPhone 8 spectral sensitivity functions. The colored bold curves, black bold curves, and broken curves represent the measured spectral sensitivities, approximations using the first component only, and approximations using the first two components, respectively.

**Figure 8 sensors-21-04985-f008:**
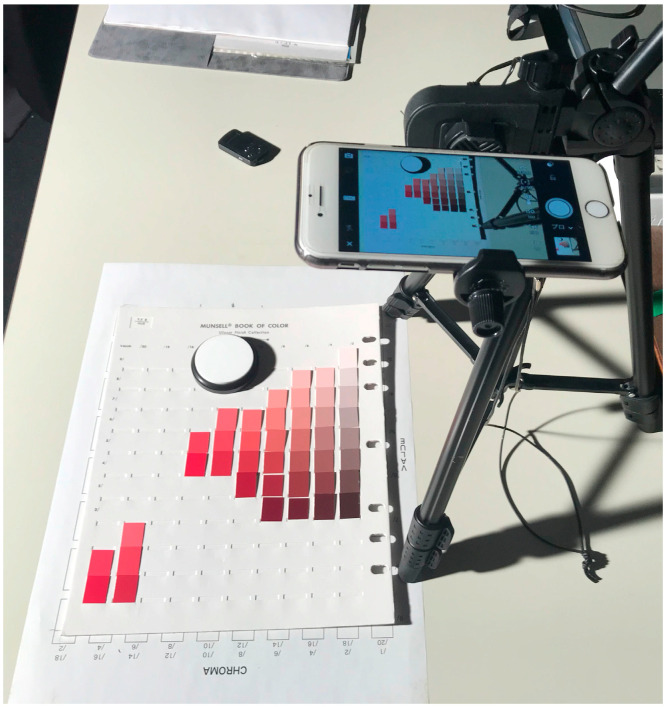
Setup for color sample imaging by a mobile phone camera.

**Figure 9 sensors-21-04985-f009:**
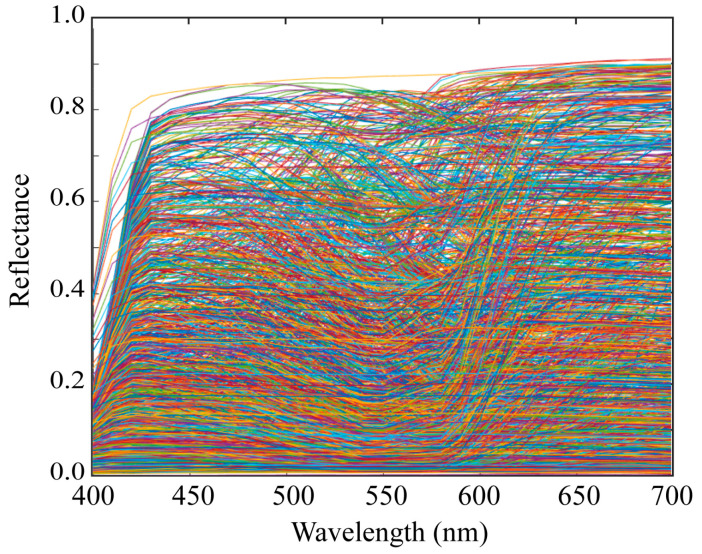
Complete set of the surface–spectral reflectances measured from all the Munsell color chips.

**Figure 10 sensors-21-04985-f010:**
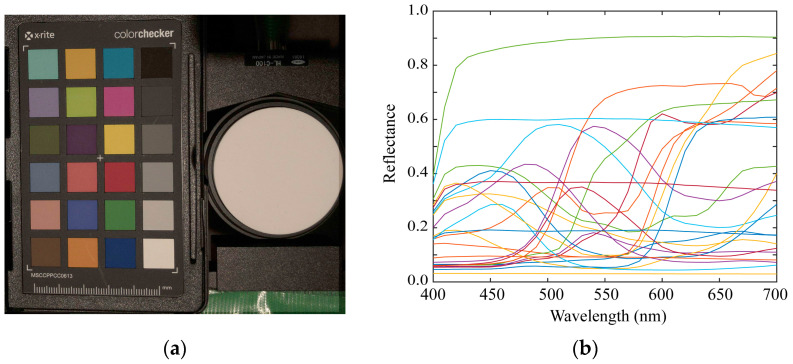
Color checkers used for reflectance estimation validation. (**a**) Imaging targets consisting of 24 color checkers and the white reference standard (Spectralon). (**b**) Spectral reflectances of the 24 color checkers measured by the spectral colorimeter.

**Figure 11 sensors-21-04985-f011:**
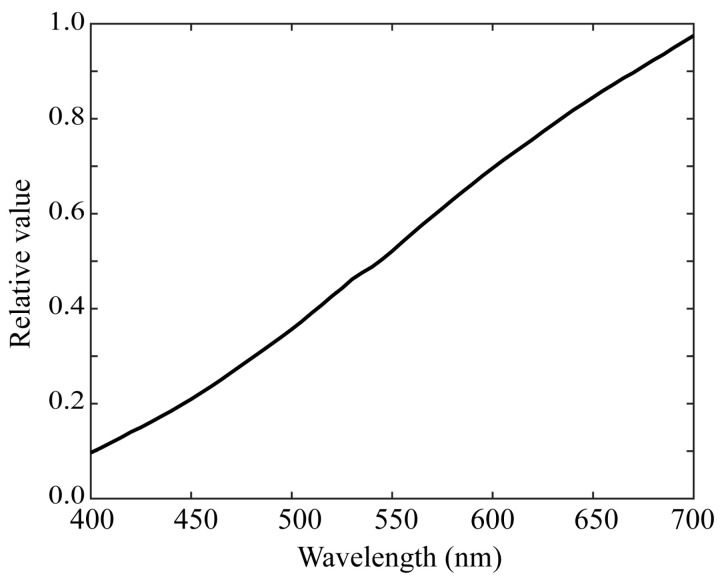
Illuminant spectral power distribution of the incandescent lamp used.

**Figure 12 sensors-21-04985-f012:**
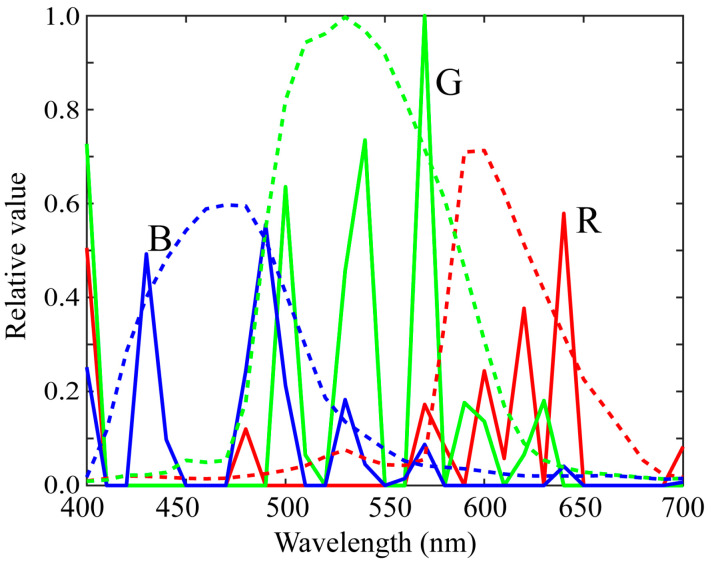
Estimation results from the normal method for the iPhone 8, where the bold curves in red, green, and blue represent the estimated spectral sensitivities for the red, green, and blue channels, respectively, and the broken curves represent the measured spectral sensitivities used as the reference data.

**Figure 13 sensors-21-04985-f013:**
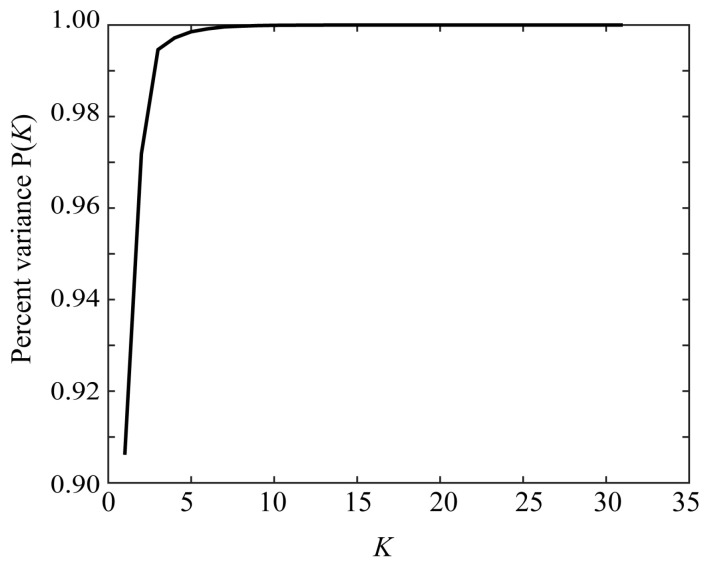
Percent variance P(K) as a function of the number of principal component *K*.

**Figure 14 sensors-21-04985-f014:**
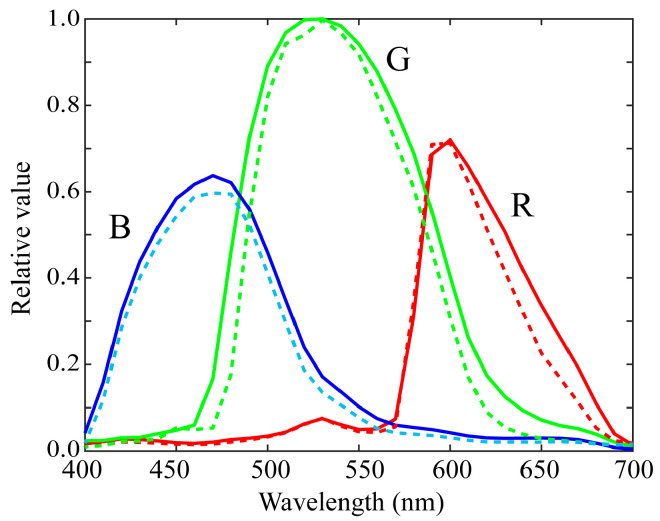
Estimated spectral sensitivity functions of iPhone 8 at *L* = 1 using all color samples, where the bold curves represent the estimated spectral sensitivities, and the broken curves represent the measured spectral sensitivities.

**Figure 15 sensors-21-04985-f015:**
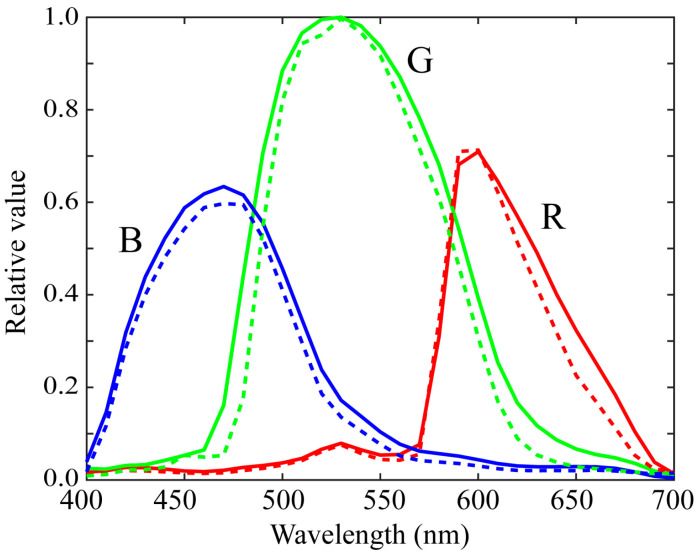
Estimated sensitivities of iPhone 8 at *L* = 1 and *m* = 10, where the bold curves represent the estimated spectral sensitivities, and the broken curves represent the measured spectral sensitivities.

**Figure 16 sensors-21-04985-f016:**
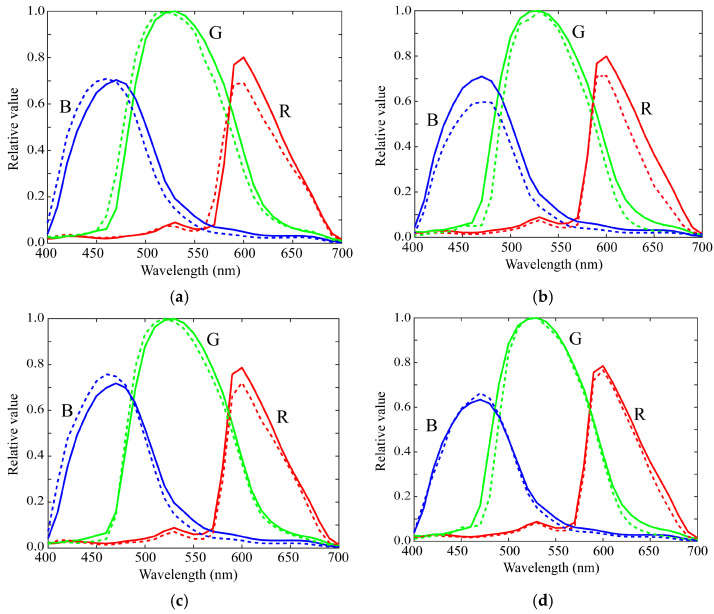
Estimated spectral sensitivities of (**a**) iPhone 6s, (**b**) iPhone 8, (**c**) P10 lite, and (**d**) Galaxy S7 edge at *L* = 1 using the 24 color checkers, where the bold and broken curves represent the estimated spectral sensitivities and the measured spectral sensitivities, respectively.

**Figure 17 sensors-21-04985-f017:**
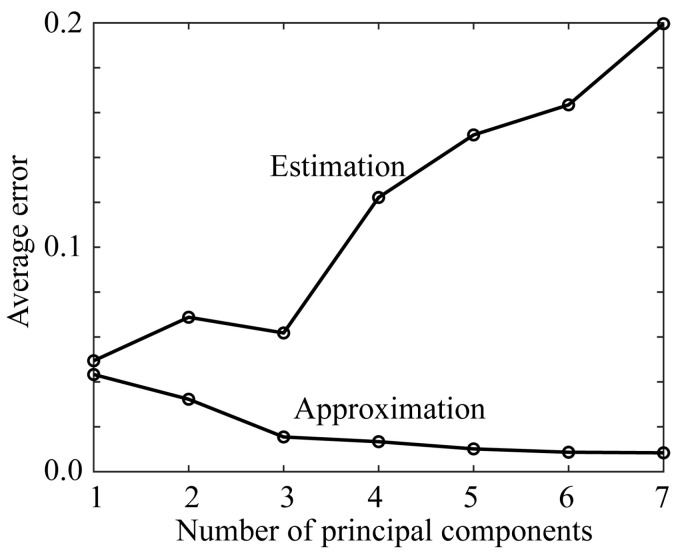
Variations in the estimation error and the approximation error as a function of the number of principal components. The error values are averaged over the four mobile phone cameras.

**Table 1 sensors-21-04985-t001:** Mobile phones measured in this study.

Manufacturer	Model	Image Sensor
Apple	iPhone 6s	Sony IMX315
Apple	iPhone SE	Sony IMX315
Apple	iPhone 8	Sony IMX315
Apple	iPhone X	Sony IMX315
Apple	iPhone 11	Sony IMX503
Apple	iPhone 12 Pro MAX	Sony IMX603
HUAWEI	P10 lite	Sony IMX214
HUAWEI	nova lite 2	Unknown
Samsung	Galaxy S7 edge	Samsung ISOCELL S5K2L1
Samsung	Galaxy S9	Samsung ISOCELL S5K2L3
Samsung	Galaxy Note10+	Samsung ISOCELL S5K2L4
Samsung	Galaxy S20	Samsung ISOCELL S5KGW2
SHARP	AQUOS sense3 lite	Unknown
SHARP	AQUOS R5G	Infineon Technologies IRS2381C
Xiaomi	Mi Mix 2s	Samsung ISOCELL S5K3M3
Xiaomi	Redmi Note 9S	Samsung ISOCELL S5KGM2
Sony	Xperia 1 II	Sony IMX557
Sony	Xperia 5 II	Sony IMX557
Fujitsu	arrows NX9	Unknown
Google	Pixel 4	Sony IMX363

**Table 2 sensors-21-04985-t002:** Average color differences between the imaged colors of the 24 color checkers captured by each camera and the simulated colors based on the measured spectral sensitivities.

Color Difference	Model			
	iPhone 6s	iPhone 8	P10 Lite	Galaxy S7 Edge
$\Delta E_{\mathrm{RGB}}$	0.01530	0.02046	0.01772	0.01689

**Table 3 sensors-21-04985-t003:** RMSE of the estimated spectral sensitivities for four mobile phone cameras at different values of *L* using the 24 color checkers.

RMSE	Model			
	iPhone 6s	iPhone 8	P10 Lite	Galaxy S7 Edge
L = 1	0.05577	0.06597	0.03841	0.03779
L = 2	0.07167	0.10714	0.05741	0.03946
L = 3	0.06378	0.07310	0.06816	0.04277

**Table 4 sensors-21-04985-t004:** Performance values in the four cases. For measurements 1 and 2, respectively, the directly measured spectral sensitivities by a monochromator in Figure 2a and the directly measured spectral sensitivities by a programmable light source in Figure 2b were used. For Estimations 1 and 2, respectively, the estimated spectral sensitivities using all color sample in Figure 14 and the estimated spectral sensitivities using only ten color samples in Figure 15 were used.

	Measurement 1	Measurement 2	Estimation 1	Estimation 2
Average RMSE	0.05241	0.05145	0.05201	0.05279
Average LAB color difference	7.055	6.082	6.973	6.749

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
