# Peer review of "Measurement and Estimation of Spectral Sensitivity Functions for Mobile Phone Cameras"

_sensors, 2021, doi:10.3390/s21154985_

Round 1

Reviewer 1 Report

The authors try to measure and estimate the spectral sensitivity functions of cameras, which is vital in the ISP algorithms development and other related applications as illustrated  in the manuscript. However, both the framework and methods are very similar to the following published paper.

Jun Jiang, Dengyu Liu, Jinwei Gu and Sabine Susstrunk. What is the Space of Spectral Sensitivity Functions for Digital Color Cameras?. IEEE Workshop on the Applications of Computer Vision (WACV), 2013.

In addition, the following issues also need to be clarified.

(1) The measured spectral sensitivities are used as ground truth in the estimated spectral sensitivities evaluation, while the measured spectral sensitivities might not be accuracy enough to reflect the spectral sensitivities of the sensors, as they are affected by many factors in the setup. It would be better to evaluate the accuracy of the measured spectral sensitivities, such as by computing the color differences between the imaged and simulated colors based on the measured spectral sensitivities.

(2) Below eq.(22), the authors say that the error was the minimum when the estimation was performed using only one principal components, which doesn't make sense and seems conflicted with the results shown in Figure 7. In practice, the spectral sensitivities might be significantly different among different sensors, the average spectral curves might not be  enough to estimate the spectral sensitivity functions.

Reviewer 2 Report

The paper aims to evaluate the spectral sensitivity of image sensor in mobile phone camera. The paper is carefully written and clear.

With respect to the paper, the following issues must be taken into consideration:

  • (minor) The paper should nominate and add in table 1 the name of the image sensor. In the vast majority of the cases, the mobile phone incorporates an image sensor produced by a different manufacturer. The paper evaluates the sensor and not the phone. Furthermore, it is not uncommon for a phone model to have multiple sensors across various production batches. While the post-processing may affect the sensitivity, the majority of information comes from the sensor.
  • (critical) In the current form, the paper has limited to no technical contribution as both the methodology and the processing are known. By simply plotting the sensitivities, the community will not benefit if this paper is published. The resolution of the figures is too small, and various phones are not identified in figures 3,4,5,9. Thus a reader will learn nothing. To me the paper may contribute to the community if the sensitivities are made public as excel data (numerical series) attached to this paper

Reviewer 3 Report

The paper "Measurement and Estimation of Spectral Sensitivity Functions for Mobile Phone Cameras" is an interesting topic that can help in estimating spectral sensitivity response functions for different mobile phones using the indirect method.

1- Lines 35 to 39 require references

2-Lines 46 to 49 require references

3-The reference [15] is an indirect method that resembles the current one. The authors in lines 76- to 78 claim that the reference [15] did not use ground truth data. But, the authors of [15] which resembles the current research justified that the ground truth or directly measured RGB spectral response functions of the smartphones are not available as the manufacturers do not share those specifications in the public domain.

The authors' contribution is limited to the creation of ground truth data to verify the indirect method results.

4-Although the authors of this research used direct measurement methods to create ground truth data, it must be compared with that of the spectral response functions (SRFs) of the smartphones from the manufacturers to verify the accuracy of the created SRFs.

5- On line 117 to 118 what is ISO and WB? Make sure that all abbreviations are explained.

6-In the Estimation of Spectral Sensitivity Functions section the equation 15 is similar to equation 21 except that it is indicated that equation 15 is the least square estimates (LSE) of equation 14 (observed outputs) while equation 21 is the LSE of weights Xr, Xg, Xb.
Please explain and clarify this ambiguity 

7-In equation 16 authors indicated that they are using three PCA vectors for each color. However, if we analyze Figures 7 b and 7c we can see that in 7b both second and third PCA vectors are similar and for 7c (blue) all principal components are the same! Why then do you have to use all three PCAs?

8-Authors created the SRF for iPhone 8! Why authors did not create the SRFs for all mobile phones listed in the table 1?

Round 2

Reviewer 2 Report

The observation from the previous round have been met, including the publication of the database.

The only suggestion, but is up to authors is to enhance the fact that the database is made public by mentioning in the abstract

Author Response

The observation from the previous round have been met, including the publication of the database.

The only suggestion, but is up to authors is to enhance the fact that the database is made public by mentioning in the abstract

>Thank you for your comments. According to the reviewer's suggestion, we have added a sentence in the abstract to inform of at the site where our database is available. 

“A variety of mobile phone cameras are measured using the system to create a database of spectral sensitivity functions, which are available at http://ohlab.kic.ac.jp/.” (Lines 24-25).

Reviewer 3 Report

The Authors' research work has been improved, but there is still one point that cannot be solved by the authors. The authors cannot get the SRFs from the mobile phone manufacturers to verify the accuracy of the created SRFs. We can disregard this issue, but the credibility and the correctness of the method are still dependable on the verification using manufacturers' data. 

The research shows the spectral sensitivities can be estimated based on one principal component as a feature extracted from the measured database. this is still a marginal contribution compared to reference [18]

Author Response

The Authors' research work has been improved, but there is still one point that cannot be solved by the authors. The authors cannot get the SRFs from the mobile phone manufacturers to verify the accuracy of the created SRFs. We can disregard this issue, but the credibility and the correctness of the method are still dependable on the verification using manufacturers' data. 

The research shows the spectral sensitivities can be estimated based on one principal component as a feature extracted from the measured database. this is still a marginal contribution compared to reference [18]

We understand your concern, and we agree that the methods' credibility can be enhanced if the manufacturers’ data is available. While the paper [18] used a compressive sensing framework to estimate the spectral sensitives, the present paper proposes an approach that combines measurement (making the database) and estimation of spectral sensitivities. Therefore, both approaches are entirely different and cannot be compared. Nevertheless, we believe that this paper adds value to many fields including imaging science and technology as the techniques to estimate spectral sensitivities of mobile phone cameras are still developing. Our study can be an effective contribution to show journal readers different approaches as there is a strong growth in the market of mobile phones.

“Even in short of the manufacturer’s data, …”  (Lines 533-534)

“We believe that this paper adds value to many fields including imaging science and technology as the techniques to estimate spectral sensitivities of mobile phone cameras are still developing. Our study can be an effective contribution to show journal readers different approaches as there is a strong growth in the market of mobile phones.” (Lines 602-606)